# Participatory Development Process of Two Human Dimension Intervention Programs to Foster Physical Fitness and Psychological Health and Well-Being in Wildland Firefighting

**DOI:** 10.3390/ijerph18137118

**Published:** 2021-07-02

**Authors:** Caleb Leduc, Sabir I. Giga, Ian J. Fletcher, Michelle Young, Sandra C. Dorman

**Affiliations:** 1Centre for Research in Occupational Safety and Health (CROSH), Laurentian University, Sudbury, ON P3E 2C6, Canada; sdorman@laurentian.ca; 2Division of Health Research, Faculty of Health and Medicine, Lancaster University, Lancaster LA1 4AT, UK; s.giga@lancaster.ac.uk (S.I.G.); i.j.fletcher@lancaster.ac.uk (I.J.F.); 3Aviation Forest Fire and Emergency Services, Ontario Ministry of Natural Resources and Forestry, Sault Ste. Marie, ON P6A 6V5, Canada; michelle.young@ontario.ca

**Keywords:** human dimensions, health, physical fitness, mental health, intervention, participatory research, psychological health, education, training

## Abstract

Intervention programs designed, delivered, and evaluated by and within organizations are a critical component in the promotion of employee health and well-being and in the prevention of occupational injury. Critical for transference of findings across complex occupational settings is a clearly articulated development process, a reliance on and evaluation of underlying theoretical foundations, and the inclusion of relevant outcomes emerging out of participatory action processes. To date, there have been no documented efforts outlining the development, implementation, or evaluation of human dimension intervention programs targeting wildland firefighters. The purpose of this paper is to outline the development of two collaborative and participatory intervention programs, targeting wildland firefighters’ physical and psychological health and well-being. Two human dimension intervention programs were developed in a collaborative, iterative and participatory process following the Context–Content–Process–Outcomes Framework. First, a physical fitness training intervention program was designed to maintain wildland firefighter’s physical fitness levels and attenuate risk of injury. Second, a psychosocial education intervention program was developed to mitigate the impact of psychosocial risk factors, foster work engagement, and decrease job stress. The current study provides evidence for the capacity of researchers and organizations to collaboratively develop practical programs primed for implementation and delivery.

## 1. Introduction

Workplaces have been identified as a priority setting for promoting physical and psychological health and well-being through interventions. Both have shown to be efficacious for improving well-being and work performance [1,2,3]. Intervention programs designed, delivered, and evaluated by and within organizations are a critical component in the promotion of employee health and well-being and in the prevention of occupational injury [4,5,6,7,8,9]. Recent efforts to enhance the methodological rigour in organizational intervention research sacrificed the ability to understand context and process, and limited the ability to learn from poorly designed or inadequately implemented interventions [10,11]. Specifically, the emphasis has errantly shifted towards appraising the effectiveness of interventions, rather than the development and implementation processes of the interventions themselves [12,13]. Critical for transference of findings across complex occupational settings is a clearly articulated development process, a reliance on and evaluation of underlying theoretical foundations, and the inclusion of relevant outcomes emerging out of participatory action processes [6,10,14,15].

Research on the design and implementation of interventions designed to foster employee well-being have yet to be synthesized for a wide range of working groups, including those in high-demand and unpredictable occupations, such as emergency response wildland firefighting. As wildland fires are a global concern and global temperatures rise, the demand for highly trained and specialized group of wildland firefighters (WFFs) also rises [16,17,18]. Aspects of the human dimensions of wildland fire include exposure to extreme physical and psychological challenges across a wildland fire season. These challenge include: rough terrain, heavy equipment, long working hours, personal risk, poor sleep, and unpredictable environmental factors, such as variations in heat; all while attempting to contain and suppress fires raging across hectares of densely forested regions [19,20,21,22,23,24,25,26]. As a result, the level of injury incidence and severity among WFFs is high, with diverse etiology and epidemiology [27,28,29,30]. Notwithstanding the aforementioned challenges, WFFs are expected to maintain a high level of physical fitness and mental acuity throughout, to ensure their ability to competently complete the task of wildland firefighting. 

### Purpose

Central to the success of any intervention program is the ability to translate theory and current evidence into practice. A primary challenge in this process is integrating the intervention in meaningful ways into the context of the organization and with the appropriate intensity to yield the desired outcomes [6,14,31]. To date, there have been no documented efforts to outline the development, implementation, or evaluation processes of human dimension intervention programs targeting wildland firefighters. Moreover, health and safety staff across this high-risk sector are routinely tasks with developing and enacting new programs within their workplace with little resources to draw upon. Therefore, the purpose of the current study is to outline the collaborative and participatory development process of two human dimension intervention programs, targeting wildland firefighters’: (i) physical fitness; and (ii) psychological health and well-being. In addition to articulating the protocol relating to the intervention programs, this article will conclude with a detailed presentation of both intervention program components and aspects of the design and implementation processes across a wildland fire season. This study adopted the intervention development framework of Karanika-Murray and Biron [11]; which proposes four elements to consider: context, content, process, and outcomes. It is hoped that through this novel application, an expansion of intervention research within wildland firefighting will result, addressing the pressing human dimension needs within wildland firefighting for the betterment of the health and well-being of its’ staff. 

## 2. Methods

### 2.1. Overall Considerations: Approach

Central to developing and implementing effective interventions in the context of the workplace is a participatory approach: wherein the end-user’s involvement is continually sought through all phases of planning, implementation, and evaluation [15,32,33]. Broadly, participatory action research is a: “collective, self-reflective inquiry; that researchers and participants undertake, so they can understand and improve upon the practice in which they participate and the situations in which they find themselves” [34]. As such, the methodology that follows outlines the design, implementation, and evaluation processes of the current intervention programs; undertaken collaboratively in an iterative process with on-going communication and feedback between: the lead author, members of the research team, and multiple levels of stakeholders within the partnering organization. During this time and across all aspects of the research project, extensive consideration was given to the context in which the interventions would be received and implemented. 

### 2.2. Intervention Setting

In the Canadian province of Ontario, WFFs are employed by the Ministry of Natural Resources and Forestry’s Aviation, Forest Fire, and Emergency Services (MNRF-AFFES) branch. The legislated wildland fire season in Ontario runs from 1 April to 31 October: each season posing unique challenges with an average of ~700 wildland fires annually, within the province’s jurisdiction of over 1,000,000 km^2^ [35,36]. Ontario’s 760 WFFs are stationed in crews of four at one of the 14 Fire Management Headquarters (FMHs) or three Attack Bases (ABs) divided into Northwestern (NWR) and Northeastern (NER) Regions [37]. FMHs and ABs work under the umbrella and operational authority of the Regional Emergency Operations Centres (REOC), where resources (personnel, aircraft and equipment) are coordinated at a regional level. During periods of escalated wildland fire activity, the Ministry Emergency Operations Centre (MEOC) establishes provincial priorities and coordinates the sharing of resources between regions. During their deployment, WFFs can be solely responsible for establishing their line camp, cooking their own meals, and sleeping in a tent.

### 2.3. Participants

The process of intervention program development, implementation, and evaluation was part of a collaborative research agreement between members of two academic institutions and a wildland fire organization. To facilitate the participatory process clear channels of communication were articulated, to ensure effective communication between both parties. This included assigning a primary point-of-contact for both groups for regular communication, and regularly scheduled team meetings with all members. Imperative for the success of this project, was the mutual recognition and respect for the unique contribution provided by each party. Organizational representation brought invaluable perspective about their: operations, practices, and programs; coupled with knowledge of their existing budgets and the demands of a wildland fire season, whereas the research team brought their expertise, from years of research within the field, and the ability to synthesize evidence from the literature to inform program content, delivery, and evaluation, consistent with sound methodological and scientific practice. 

#### 2.3.1. Academic Team

Representation from the academic institution partners included the lead author who at the time was both a doctoral student and research associate with a health and safety research centre. Further, academic partners included the Director from an occupational health and safety research centre within a Canadian University and collaborating faculty from both domestic and international universities, providing expertise in content areas and intervention development methodology. The academic team received additional support from technical staff within the health and safety research centre. 

#### 2.3.2. Organization Team

Representation from the partnering fire organization included the Fire Science and Technology Program Leader and their dedicated, Health and Wellness Specialist. The Program Leader provided insight to and coordination with provincial priorities and the allocation of resources. The Program Lead also led strategic priority setting and process/outcomes review. The Health and Wellness Specialist, a Registered Kinesiologist and Certified Athletic Therapist, provided critical input to intervention material development, implementation, and evaluation processes in addition to supporting, scheduling, and facilitating data collection procedures. Input from members of local and senior management teams provided leadership and guidance throughout the process and championed participation across the organization. Finally, the Health and Wellness Specialist coordinated feedback on various stages of the development process from a minimum of five wildland firefighter staff members. 

### 2.4. Preliminary Considerations: Theoretical Perspective

From the outset of the development process, all parties agreed to the adoption of the Job Demands-Resources (JD-R) Theory to guide the understanding and to frame the perspective of wildland firefighting tasks as either: a demand, or a resource. The JD-R model has been applied extensively in empirical research and utilized across a range of organizations around the world [38,39,40]; evolving into a mature theory expounding on the relationships between job characteristics and employee well-being [38,41,42,43,44]. Central to the JD-R Theory’s widespread acceptance is an inherent flexibility when applied to various occupational settings. The structure assists in detecting and understanding antecedents of employee well-being [39]. Finally, the JD-R Theory has served as foundational for a burgeoning field of intervention research targeting individual components within it, while also contributing to the evaluation of the theory as a whole [38,39,45,46].

### 2.5. Procedure

Primarily, the Organization’s Health and Wellness Specialist and Fire Science and Technology Program Lead worked in close collaboration with the lead author and Director from the university’s health and safety research centre. The initial high-level, project conceptualization meeting took place in July 2015 where: physical fitness, and psychosocial risk factors; emerged as priority areas for intervention. Representation from the Organisation communicated a need to support WFFs, both physically and psychologically, to meet the demands of a wildland fire season. Through consultation with the Health and Wellness Specialist and Program Lead, it was agreed that two independent intervention programs would be developed to address each human dimension of wildland fire, separately. Specifically, the Organization identified an opportunity to build upon and examine the effectiveness of an established physical fitness training program; while internally there was a strong desire to begin addressing psychosocial risk factors. 

Leveraging the cyclical nature of wildland fire seasons, the majority of the development work took place between active fire seasons. From September 2015 through February 2016, several in-person and conference-call team meetings were coordinated to provide input on the development of both intervention programs’ material, content, and delivery. These meetings included representation from: local, regional, and senior management; fire and operations staff; health and safety personnel; and wildland firefighters; all of whom, drew on existing programming, policies and procedures, in addition to field experiences, to provide input. During the Spring of 2016, presentations to and consultation with both Regional Management Groups occurred to allow final program revisions from all potential, participating locations. The development of each intervention program (physical and psychological) followed the four dimensions: context, content, process, outcomes and detailed as follows [11].

### 2.6. Context: Tailoring to Wildland Firefighting within the Organization

Context refers to an understanding of the environment in which the intervention will occur and its potential to impact outcomes [11]. To ensure good contextual understanding, the research team relied on consultations with both upper and local levels of management, and multiple site visits to various locations across the organization, to gain a thorough understanding of the environment in which any intervention would be delivered. Moreover, members of the organization relied on the research team to appraise and synthesize the current literature around the topic areas, to inform the development of evidence-based content for the intervention programs; addressing the physical and psychological demands of wildland firefighting. 

### 2.7. Content: Alignment with Existing Resources in Wildland Fire

Content refers to the substance or material-content of the intervention [11]. To provide relevant content, due consideration was given to develop workshop and training materials, that they were empirically driven, but also to ensure that they were presented in a way that was visually appealing and accessible to wildland firefighters [11]. Robson et al.’s [47] training model was used to guide consultations between members of the research team and the organization. Here, members of the organization provided perspective regarding existing programs internally, as they related to either the physical or psychological demands of wildland firefighting. In particular, the organization sought to identify opportunities for leveraging successful components of previously piloted initiatives or existing policies. Members of the research team brought knowledge of best practices pertaining to the subject matters, and promising practices from the scientific literature. 

### 2.8. Process: Ensuring Informed Delivery

The third element: process, refers to the manner in which the intervention was delivered and received by wildland firefighters [11]. Stakeholder consultation in this regard was critical: informing the development of intervention procedures, with consideration given to fire response, personnel availability, and access to wildland firefighters themselves. From the research methodology perspective, members of the research team guided conversations around intervention fidelity and consistency, assessing dose response, and ensuring reliability and replicability of procedures across multiple locations. 

### 2.9. Outcomes: Determining Evaluation Considerations

Finally, outcome refers to considerations with respect to the measurement and evaluation of the effects of the intervention [11]. The consideration and selection of relevant measures, used to evaluate both the intervention and contribute to theory, was a mutual process with endorsement by several levels of management, wildland firefighters and the research team. Critical from the perspective of the organization was the selection of factors that would inform subsequent program development and evaluation and contribute to existing organizational health and safety objectives and outcomes. Driving decision making on the part of the research team was the structured identification of measures that would allow for subsequent, theoretical contributions to the scientific literature, in addition to a comprehensive evaluation of how and whether the intervention programs were effective. 

## 3. Results

### 3.1. Development

Building off the recommendation from Gilbert et al. [48] for “deploying programs that target resources specific to task demands” (p. 10), and leveraging existing wellness programs and initiatives where possible, two resource-building intervention programs were developed; guided by participatory action research principles [15,34]. A physical fitness training intervention was designed to: maintain wildland firefighter’s physical fitness level; and a psychosocial education intervention program was developed to mitigate the impact of psychosocial risk factors and foster work engagement. Both interventions had secondary aims to attenuate the risk of injury. What follows, is the detailed description of (i) key contextual considerations, (ii) the content of the interventions that were developed, (iii) the critical elements of implementation process, and (iv) evaluation measures; for the physical fitness training and psychosocial educational interventions. 

### 3.2. Understanding Context: Physical Job Demands in Wildland Fire

An extreme occupation, wildland firefighting presents employees with arduous physical and psychological demands, including rough terrain, heavy equipment, long working hours, personal risk, poor sleep and a variety of unpredictable environmental factors (e.g., weather, wildlife) all while attempting to contain and suppress wildland fires raging across hectares of densely forested regions in extreme heat [19,20,21,22,24,26,49]. Research over the past 15 years has consistently estimated the daily energy demands whilst fighting wildland fires to exceed 4500 kcal, a result of navigating rough terrain while carrying or pulling heavy equipment, such as pumps and hoses [22,50,51,52]. Additional research has found that a WFF’s average heart rate over the course of a shift ranged from 110 to 160 beats per minute, reflective of the variation in intensity required by various duties throughout the day [22,53]. In terms of metabolic equivalents (METs), wildland firefighting has been shown to average 6.5 METs over the course of a day, with an upper threshold of 9 METs during peak exertion [54]. Meeting these demands requires above average fitness levels [55], with many additional factors to consider including: hydration [56,57,58], nutrition [52,59] and thermoregulation across variable thermal environments [23,26,53].

In order to be employed as a wildland firefighter in Canada, individuals must successfully complete the Canadian Physical Performance Exchange Standard for Type 1 Wildland Firefighters, also known as the WFX-FIT [60]. The WFX-FIT was implemented in 2012 as a “valid, job-related physical performance standard used to determine whether an individual possesses the physical capabilities necessary to meet the rigorous demands encountered while fighting wildland fires” [60]. Successful completion of the WFX-FIT test is valid for a period of 90 days that immediately precedes the beginning of each wildland fire season. However, once the fire season begins, limited research has evaluated the physical fitness levels of WFFs [54] as there are currently no minimum standards or requirements that they must maintain across the fire season. 

The physically demanding nature of wildland firefighting commonly manifests itself in observations of injury trends among WFFs. Within the partnering organization, sprains and strains (45.8%) and contusions and wounds (25.5%) are the most frequently reported types of injuries [61]. These findings are consistent with recent analyses of injury patterns reported by WFFs across the United States [27,29,62]. Moreover, the MNRF-AFFES has identified both the frequency and severity of lost-time claims as a result of musculoskeletal injuries as an area of focus for prevention, necessitating a more proactive approach to maintaining task-specific physical fitness across the duration of a wildland fire season [63].

### 3.3. Understanding Context: Psychological Job Demands in Wildland Fire

Exposure to the recurring uncertainty associated with fighting wildland fires results in significant occupational and environmental stressors. Despite this, limited research has evaluated the psychological demands and subsequent psychological well-being of WFFs [64]. Gordon and Larivière [24] found that nearly half of surveyed WFFs in Ontario self-reported high levels of job stress over the course of a fire season. A subsequent study of a smaller sample of WFFs in Ontario found that experiences of overall job stress increased from mid- to post-season, though scores remained within limits indicating that perceived work stress was comparable to the average range in normative data, for workers employed in the skilled-maintenance sector [65,66,67]. This same study identified ‘perceived level of organizational support’ as the primary driver of WFF’s overall job stress score, which increased significantly over the course of the fire season [66]. To this point, no comprehensive evaluation of psychosocial risk factors had been completed in the context of wildland fire, nor did the organization possess complete data on the impact of its psychosocial climate on organization-level outcomes or lost-time claims within their workforce.

### 3.4. Content: Physical Fitness Training Intervention

Physical fitness refers to the ability of body systems to function in a coordinated and efficient manner to perform routine activities of daily living, including work tasks [68,69]. With regard to in-season resources available for WFFs, Ontario’s MNRF-AFFES developed a ‘Commit to be Fit’ task team in 2013 to guide the development of a physical fitness program. From 2013 to 2015, several iterations of the fitness program were developed, piloted and evaluated internally based on feedback from all levels of staff and management across the organization [37]. The goal of the MNRF-AFFES’ Commit to be Fit program is to “build and maintain strength, flexibility and endurance, and maintain mental alertness” [37]. MNRF-AFFES recognizes WFFs as ‘occupational athletes’, and therefore permits them to engage in physical activity for a period of up to one hour within the first two hours of work when at base. The Commit to be Fit program also identified a ‘fitness lead’ at each location to advocate for participation and assist in establishing a culture of fitness and well-being. Resources were made available to the fitness leads and WFFs at each location including an exercise library and support for purchasing training equipment [37,63]. While the program has been well received among WFFs, several challenges remained including: participation, availability of equipment, timing, management support, and training structure [63]. In addition, given the cost of the program, there was interest in confirming whether value was added organizationally. To date, no formal evaluation of program participation, or evaluation of efficacy via established fitness tests had been completed.

As such, the development of the physical fitness training intervention program content was designed to provide structure to the existing Commit to be Fit program. Specific to informing the content of the physical fitness training intervention program, feedback was collated from five wildland firefighters and local management team members by the Health and Wellness Specialist, who worked directly with the research team to develop the physical fitness training intervention for the subsequent fire season. The design process included several consultations with various levels of management across the organization with the overall aim of enhancing and evaluating the Commit to be Fit program by formalizing a training structure, providing accountability, and educating wildland firefighters with regard to task-specific physical fitness principles. The physical fitness training intervention contained five elements: (1) Educational workshop; (2) Formalized training schedule; (3) Logging system; (4) Wearable fitness tracker; and (5) Personalized feedback (see Figure 1). 

#### 3.4.1. Educational Workshop

A 30-min, educational workshop was developed and presented to wildland firefighters as ‘occupational athletes.’ The workshop was illustrated with examples of the physical demands required to perform their routine tasks and stressed the importance of maintaining a high level of physical fitness. Energy systems (e.g., anaerobic vs. aerobic) and basic training principles (e.g., specificity, periodization, variation, and maintenance) were also covered in addition to an overview of the remaining four elements of the intervention. 

#### 3.4.2. Training Schedule

A formalized training schedule was developed for the wildland firefighters to follow over the course of the fire season. It was communicated to participants that participation in the exercise program was expected on days when wildland firefighters were stationed at their home base, but not on active deployment. The schedule was designed to encourage variation in activity, rotating wildland firefighters through: cardiovascular fitness, plyometric training, weight training, and active rest days. The schedule was also developed collaboratively between the organization’s Health and Wellness Specialist with input from WFFs and the researcher based on availability of equipment at each base and in an effort to allow for efficient participation (within the 1-h permitted). 

#### 3.4.3. Logging System

A system for logging participation, either electronically or in paper formats, in the fitness training room, was developed as an accountability and motivation tool for participants. 

#### 3.4.4. Wearable Fitness Tracker

Wildland firefighters were provided with a wrist-worn fitness tracker to encourage on-going awareness of and participation in training activities. Data from fitness trackers were not requested by either the research team or the organization; as they were provided as an incentive to engage and support participation in the physical fitness program.

#### 3.4.5. Personalized Feedback & Training Support

Personal feedback from the initial pre-season fitness measures session was provided to each of the participating wildland firefighter, allowing for an understanding of their relative strengths and weaknesses compared to the provincial average, in addition to general population and elite-level athlete normative data. 

### 3.5. Content: Psychosocial Education Intervention

With regard to the psychological safety and well-being of WFFs over the course of a wildland fire season, it was found that several reactionary supports existed. On an organization-wide level, as public service employees, WFFs have access to an Employee and Family Assistance Program and a comprehensive Workplace Discrimination and Harassment Program. Further, and internal to the MNRF-AFFES, WFFs can access a peer support program if experiencing psychological distress; particularly in response to critical incidents. Locally, an ad hoc health-promoting committee leads holistic wellness initiatives at a local level across all work locations. However, evaluation of program effectiveness and documentation of participation in the aforementioned programs does not exist. Moreover, there is no published evidence of a program, designed to educate WFFs on workplace issues impacting their psychological safety and well-being. As such, given the extremely high psychological job demands, wildland firefighting presented a unique opportunity for evaluating the efficacy of proactive, resource-based educational intervention program. 

Therefore, a psychosocial education intervention program was designed as a new initiative within the organization aimed at improving WFFs knowledge and understanding of psychosocial risk factors, both in general, and then contextually in wildland fire. Further, the intervention aimed at educating and informing WFFs of the support systems and resources that are already accessible to them over the course of a wildland fire season. In collaboration with the partnering organization’s management, Health and Wellness Specialists, and wildland firefighters, a series of educational fact sheets pertaining to psychosocial risk factors were developed. The topics and content for each of the fact sheets were derived from the Guarding Minds at Work [70] suite of resources that have emerged from the development of a National Standard of Canada ‘Psychological Health and Safety in the Workplace- Prevention, Promotion, and Guidance to Staged Implementation’ (CAN/CSA-Z1003-13/BNQ 9700-803/2013).

The psychosocial education intervention had two primary components (see Figure 2). First, a 45-min workshop was developed to provide an overview of psychosocial risk factors in general and then specifically describing the 13 factors in the context of the organization and wildland firefighting. The second component was the development of 13, one-page fact sheets, each highlighting one psychosocial risk factor. Each fact sheet followed a consistent format and was divided into three sections: (i) an overview of the risk factor in the context of wildland firefighting; (ii) a discussion on its relevance to wildland firefighting; and (iii) an overview of psychosocial risk factors in general. Additionally, present within each fact sheet were images taken directly from the organization’s media library to complement the topic material. The topics were as follows: civility and respect, psychological job demands, work–life balance, psychological support, organizational culture, clear leadership and expectations, growth and development, recognition and reward, involvement and influence, workload management, engagement, psychological protection, and protection of physical safety.

### 3.6. Process: Implementing the Physical Fitness Training Intervention

Both intervention programs were designed to be delivered and contained entirely within and across one wildland fire season. Specific consideration was given to each component of both intervention programs and are outlined as follows: 

#### 3.6.1. Educational Workshop

The 30-min, in-person, educational workshop was designed to be delivered jointly by the researcher and the organization’s Health and Wellness Specialist in a group setting at each participating location at the outset of the fire season. The session allowed for the researcher, as the subject matter expert, to answer questions relating to design; and to ensure consistency across all participating work locations. The Health and Wellness Specialist provided the link to organizational objectives and was able to articulate the commitment and involvement of the organization alongside the intervention materials. 

#### 3.6.2. Training Schedule

The training schedule was created and shared with WFFs with the expectation of adherence on all days when WFFs were stationed at their home base; but not when they were on active (i.e., fire) deployment. The schedule was placed in the fitness area of each base and available through the organization’s intranet. Orientation to the training schedule was provided by the organization’s Health and Wellness Specialist alongside the fitness lead, at each work location, at the outset of the fire season and subsequent to the educational workshop. 

#### 3.6.3. Logging System

An orientation to the logging system was provided by a member of the research team alongside the organization’s Health and Wellness Specialist. Both modalities for completing the logging activity were provided within the fitness area to facilitate completion. A touchscreen tablet was placed in a locked floor stand in the training room at each participating location with a workout log survey, preloaded and utilizing a free offline and secure application. WFFs were also permitted to complete the workout log in paper format and place the completed log in a locked box adjacent to the tablet. Logging method was decided by the participant, according to their preference.

#### 3.6.4. Wearable Fitness Tracker

Wearable fitness trackers, especially within the context of organization-wide implementation have proven effective over time at increasing participation in aerobic activity [71]. The fitness trackers were provided to the wildland firefighters at the beginning of the season, oriented to the corresponding smartphone application, and WFF were encouraged to use them throughout the wildland fire season to support, monitor and track their activity levels at their discretion. 

#### 3.6.5. Personalized Feedback & Training Support

The personalized feedback report was provided to each WFF via email within 2 weeks following the completion of baseline fitness measures (see outcomes section, to follow). Further, the organization’s Health and Wellness Specialist visited each participating location twice throughout the wildland fire season, to provide support to the wildland firefighters and reinforce each of the four previous elements of the intervention. Additionally, the Health and Wellness Specialist was able to respond to questions and issues that arose throughout the season, serving as a knowledge resource and subject matter expert for explanations of an individual’s feedback and demonstrations of exercises and equipment.

### 3.7. Process: Implementing Psychosocial Education Intervention

This 45-min workshop was delivered in-person and within a group setting at each work location, jointly, by a member of the research team (as subject matter expert) and supported by the organization’s Health and Wellness Specialist. The fact sheets were designed to support and reinforce the initial workshop in two ways: (1) individually to each WFF via email, and (2) in a group setting posted in at least two common areas around their work location in an 11” by 17” poster-size format. 

### 3.8. Outcomes: Selecting Meaningful Measures

The selection of outcomes was designed to: complement existing knowledge pertaining to wildland firefighter health and well-being, supplement organizational objectives and program evaluations, and make a unique contribution to scientific literature and intervention theory research. To this end, a cluster, randomised control trial methodology with pre- and post-intervention measurement points was selected to evaluate the implementation of both intervention programs with participating work locations across a wildland fire season. The RE-AIM Framework was selected to guide evaluation considerations beyond effectiveness and respond to calls to enhance our understanding of how and why interventions may be effective [46,72,73]. The RE-AIM Framework distributes criteria to be evaluated across five dimensions: reach, effectiveness, adoption, implementation, and maintenance [73]. Aspects of intervention implementation reflective of intervention fidelity and dose delivered were built into the delivery and engagement components of the intervention programs themselves. For instance, the logging system for the physical fitness training intervention provides a mechanism for assessing workout activity, while the provision of the psychoeducational material electronically by email allowed for verification of delivery and read receipts. Paramount to the successful evaluation of both intervention programs was the completion of primary and secondary outcome measures both pre- and post-intervention delivery, within the confines of a single fire season. 

### 3.9. Primary Outcome Measures

Selected measures pertaining to physical fitness completed pre- and post-intervention to evaluate the effectiveness of the intervention programs included: anthropometrics (e.g., height, weight, BMI), anaerobic capacity, grip strength, flexibility, and core muscle strength. Survey measures to assess 13 psychosocial risk factors and psychological capital were also selected. 

### 3.10. Secondary Outcome Measures

As a secondary outcome, work engagement was measured pre- and post-intervention. Due to the cyclical nature of the fire season, job stress was measured only measured post-intervention at the end of the fire season given that the majority of wildland firefighters are only employed seasonally with the organization and would have no prior experience to draw on at the pre-intervention measurement point. Finally, incidence of injury across the wildland fire season was to be recorded as an indicator of intervention effectiveness. 

## 4. Discussion

The central aim of the present study was to articulate the collaborative and participatory development process of two human dimension intervention programs within wildland firefighting to promote physical fitness and psychological well-being. Guided by the Context–Content–Process–Outcomes framework proposed by Karanika-Murray and Biron [11], and through the application of participatory action research principles, the current study provides evidence for the capacity of researchers and organizations to collaboratively develop practical programs primed for implementation and delivery. Moreover, the detailed presentation of both intervention program components and implementation strategy, makes several key contributions to our understanding of intervention research, especially within the context of wildland firefighting which is a growing global phenomenon.

First, the project lends credence to the importance of adopting a participatory approach throughout the entirety of the research process [15,32,33,74]. This approach is critical for the long-term sustainability of an intervention program; as it successfully aligns existing programs with the intervention and does not create processes that will not be sustainable in the absence of researcher involvement or facilitation. Second, a thorough understanding and level of embeddedness within the culture and context of the workplace is key to both designing intervention program material, but also in creating and selecting evaluation measures and processes that are meaningful to both the organization and academic communities [6,10,75]. Third, the design of intervention content needs to be completed in light of, and in a complementary fashion to, existing resources, policies, or programs. In this instance, the content of the physical fitness training intervention has provided structure to an existing organizational initiative, ‘Commit to be Fit’, whereas the psychosocial education intervention was a new initiative within the organization to fill an identified need. For both, the content of the interventions was designed to reach wildland firefighters: visually and personally, through the substance of the material, with pertinent examples relevant to the completion of their regularly assigned tasks. Finally, and with regard to the process of implementation and evaluation of the intervention programs, embeddedness within existing policies and practices, while leveraging internal expertise from all levels of management, was vital [46,76].

## 5. Conclusions

The current human dimension intervention research is not without its limitations. First, it is acknowledged that the process was undertaken between established collaborators who engaged in the development process building on nearly a decade of productive partnership. As such, the mutual trust and familiarity between research and organizational agents may be viewed as a confounder to the development process. To that end, it is acknowledged that the current research represents development processes for one wildland fire organization and may not be representative of the needs, capacity and objectives of agencies across different jurisdictions.

Second, the current development process was governed in several capacities by budgetary, seasonal and practical considerations. The capacity of the research team was dictated in large part by funding received to support the project and the confines of a doctoral research project; which was involved in subsequent phases of implementation and evaluation of the human dimension interventions. Limitations, from the perspective of the organization, included the capacity and availability of support personnel, and the challenges associated with implementation within an active wildland fire response season.

Future research should continue to objectively and transparently articulate the development process of human dimension interventions, both in terms of replicability and to encourage a process of continual improvement and learning across other jurisdictions. There is merit to the transparent documentation of understanding the lessons learned through the development processes. Several aspects of generalisability are worth noting. Given wildland fires are becoming an increasingly global concern and the demand placed on highly trained and specialized group of wildland firefighters (WFFs) continues to rise, the current articulation of the participatory development process guided by the context, content, process, and outcomes framework provides an example of the elements that future research should consider during the planning phases of other human dimension interventions within wildland fire; including: smoke exposure, heat stress, or fatigue. Moreover, and with regard to generalisability of the approach to other sectors and contexts, it is anticipated that researchers and organisations across other high-risk (e.g., firefighters), seasonal (e.g., treeplanters, farming labourers) or dynamic (e.g., first responders) occupation groups would also benefit from a collaborative and participatory intervention program development process. Further, it is anticipated that the ability of the current process to develop intervention programs that address salient research and organisational priorities provides evidence for adoption across other physically demanding occupation groups including mining, forestry construction. Finally, subsequent research should continue to evaluate the effectiveness of participatory human dimension interventions and initiatives across high-risk occupation groups and wildland firefighting and in particular for their ability to positively influence mutually desirable outcomes.

## Figures and Tables

**Figure 1 ijerph-18-07118-f001:**
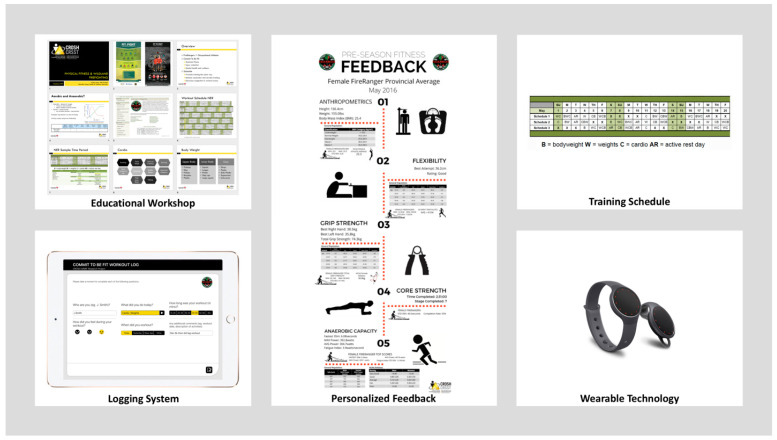
Fitness training intervention components.

**Figure 2 ijerph-18-07118-f002:**
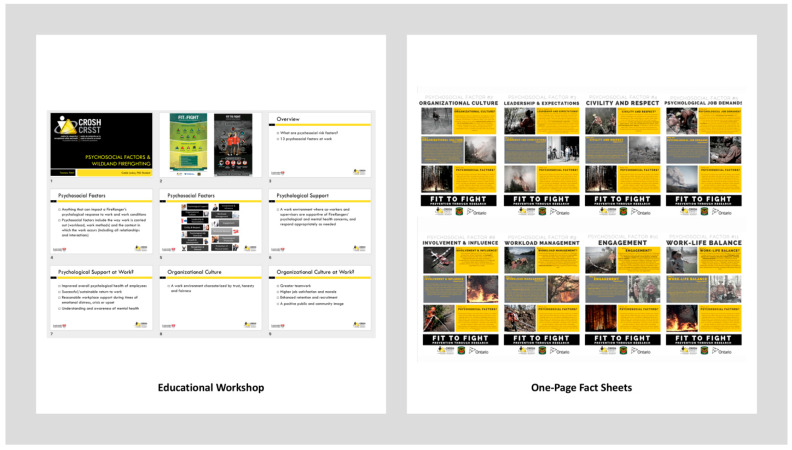
Psychosocial education intervention components.

## Data Availability

No new data were created or analysed in this study. Data sharing is not applicable to this article.

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
