# Peer review of "Participatory Development Process of Two Human Dimension Intervention Programs to Foster Physical Fitness and Psychological Health and Well-Being in Wildland Firefighting"

_ijerph, 2021, doi:10.3390/ijerph18137118_

Round 1

Reviewer 1 Report

The title talks about physical fitness, but in the development it says that a physical training intervention was designed to maintain the level of physical fitness, so it should be clarified: fitness or physical fitness?
A health specialist is mentioned, but it is not indicated which qualification he/she has, since it seems that he/she is the one who has designed the training programme, does this person have the appropriate qualification to be able to determine the training system?  
In relation to the sample, they indicate the number of academics and the organising team, but I could not find the number of participants who were WFFs.
I am missing the levels from which the participants start, from the physical and psychological aspect, I understand that when passing the WFX-FIT tests, all participants had a certain degree of physical condition. What was it at the end of the research? As for the psychosocial demands, they are based on previous studies, which is correct, but what initial stress levels did the sample subjects start with? What stress levels did they have at the end of the research? 
The section on evaluation is very rambling, it talks about many organisational things, but does not specify what tests were carried out to check the progress. I find it strange that by telling the participants that they were expected to participate in the exercise programme on the days they were at their home base, that they all attended, a circumstance that is not indicated, as the number of the sample is unknown and therefore the % follow-up cannot be determined.

Author Response

Point 1: The title talks about physical fitness, but in the development it says that a physical training intervention was designed to maintain the level of physical fitness, so it should be clarified: fitness or physical fitness?

Response 1: This is an important point to clarify and appreciated. We have added a definition of physical fitness (Line 341-342) and revised throughout the entire manuscript for consistent use of the term ‘physical fitness’, where appropriate, in lieu of only ‘fitness’. 

Point 2: A health specialist is mentioned, but it is not indicated which qualification he/she has, since it seems that he/she is the one who has designed the training programme, does this person have the appropriate qualification to be able to determine the training system?

Response 2: This remark is insightful as the health and wellness specialists’ qualifications were central to their ability to contribute to the content of the physical fitness training programme. Their credentials, which include being a Registered Kinesiologist and a Certified Athletic Therapist were added on Line 166.

Point 3: In relation to the sample, they indicate the number of academics and the organising team, but I could not find the number of participants who were WFFs.

Response 3: This is a valid point and varied across the different components of the interventions. After reviewing our records, we are confident that a minimum of five wildland firefighters were consulted on each of the intervention programmes and have added statements to that effect on Line 172 and Line 364.

Point 4: I am missing the levels from which the participants start, from the physical and psychological aspect, I understand that when passing the WFX-FIT tests, all participants had a certain degree of physical condition. What was it at the end of the research? As for the psychosocial demands, they are based on previous studies, which is correct, but what initial stress levels did the sample subjects start with? What stress levels did they have at the end of the research? The section on evaluation is very rambling, it talks about many organisational things, but does not specify what tests were carried out to check the progress.

Response 4: Given that the current paper only aims to describe the process through which the interventions and associated methodology for implementation and evaluation were developed, no results are presented with regard to any of the selected outcome measures. The research team worked extensively to draw on what we already knew from other studies and existing policy and procedures to understanding contextually where the WFFs would be at when starting on the fire season, but had no point of reference for comparing where they might have been at the end of a typical fire season. As such, the development of this intervention with both pre- and post-intervention measurement points is unique in its potential contribution to fill that current gap in the literature. Clarity to this end was added from Line 535 to Line 546 in our description of the selected primary and secondary outcome measures.

Point 5: I find it strange that by telling the participants that they were expected to participate in the exercise programme on the days they were at their home base, that they all attended, a circumstance that is not indicated, as the number of the sample is unknown and therefore the % follow-up cannot be determined.

Response 5: This is a valid point, and the authors acknowledge we neglected to highlight the process evaluation mechanisms through which aspects of implementation would be monitored were actually built into the intervention components themselves. This was addressed in Line 526 through 532, where the logging system (physical fitness intervention) and email confirmations (psychosocial education intervention) components of the interventions facilitated the process evaluation aspects of fidelity and dose delivered.

Reviewer 2 Report

This study has articulated the collaborative and participatory development process of two human dimension intervention programs within wildland firefighting to promote physical fitness and psychological well-being. The study provided evidence for the capacity of researchers and organizations to collaboratively develop practical programs for implementation and delivery. In addition, the researchers here have presented the intervention program components and the implementation strategy. These could demonstrate key contributions to the understanding of intervention research, especially within the context of wildland firefighting. Since this aspect has no border, the findings from this research can be broadly taken into account in other regions of the world. That would be desirable to read few lines on the generalisability of these findings here and if and how these can be taken by others around the globe, considering some fundamental differences against some common interests. The authors have also mentioned the limitations of the study and how future research can be taken the steps forward. 

Author Response

Point 1: This study has articulated the collaborative and participatory development process of two human dimension intervention programs within wildland firefighting to promote physical fitness and psychological well-being. The study provided evidence for the capacity of researchers and organizations to collaboratively develop practical programs for implementation and delivery. In addition, the researchers here have presented the intervention program components and the implementation strategy. These could demonstrate key contributions to the understanding of intervention research, especially within the context of wildland firefighting.

Response 1: This comment is well taken and greatly appreciated! We are thankful that the reviewer has seen the strengths and unique contributions of our intervention study in this high-risk and under-studied occupational context.

Point 2: Since this aspect has no border, the findings from this research can be broadly taken into account in other regions of the world. That would be desirable to read few lines on the generalisability of these findings here and if and how these can be taken by others around the globe, considering some fundamental differences against some common interests.

Response 2: This point is noted, and a few sentences have been added from Line 611 to Line 620 to address the generalizability not only within wildland firefighting, but other high-risk, dynamic or seasonal occupation groups.

Point 3: The authors have also mentioned the limitations of the study and how future research can be taken the steps forward.

Response 3: This point is appreciated, thank you!

Reviewer 3 Report

In the present study, a fitness training intervention was designed to maintain, not improve, wildland firefighter’s fitness level along with a psychosocial education intervention program aimed to mitigate the impact of psychosocial risk factors and foster work engagement.

However, the present study is instead a narrative description of events with no quantitative or qualitative data reflecting the study’s endpoints before and after the interventions. Yet, although one would have thought that the proposed interventions would be aimed to attenuate collectively, for instance, the risk of self-injury, limit fire-related damages as primary study-endpoints among other things, the authors chose a somewhat unclear meaning of their study, i.e. “to articulate the collaborative and participatory development process of two human dimension intervention programs within wildland firefighting to promote physical fitness and psychological well-being".

Author Response

Point 1: In the present study, a fitness training intervention was designed to maintain, not improve, wildland firefighter’s fitness level along with a psychosocial education intervention program aimed to mitigate the impact of psychosocial risk factors and foster work engagement.

Response 1: The reviewer is correct, in that the primary aim of this manuscript is to document the development process through which two novel intervention programmes were developed.

Point 2: However, the present study is instead a narrative description of events with no quantitative or qualitative data reflecting the study’s endpoints before and after the interventions. Yet, although one would have thought that the proposed interventions would be aimed to attenuate collectively, for instance, the risk of self-injury, limit fire-related damages as primary study-endpoints among other things, the authors chose a somewhat unclear meaning of their study, i.e. “to articulate the collaborative and participatory development process of two human dimension intervention programs within wildland firefighting to promote physical fitness and psychological well-being".

Response 2: The aim of the current manuscript, submitted to this special issue of the IJERPH, is focused solely on articulating the collaborative and participatory processes of the two intervention programs, the result of which is the detailed presentation of the intervention development itself. This fills a gap in both the workplace intervention research (Line 90 to Line 92), and a practical need for health and safety staff in the sector tasked with enacting health promotion programming (Line 92 to Line 94). The study aims to provide evidence for how the field of workplace health interventions is moving forward, highlighting the capacity and necessity of researchers and organizations to collaboratively develop practical programs for implementation and delivery in an understudied and increasingly relevant and vital occupation group with implications across a number of other high-risk, dynamic or safety sensitive sectors (Line 609 to 618).

Round 2

Reviewer 1 Report

All queries have been satisfactorily answered.

Author Response

We would like to thank Reviewer 1 for their time and expertise.